# A Framework to Develop Interventions to Address Labor Exploitation and Trafficking: Integration of Behavioral and Decision Science within a Case Study of Day Laborers

**Matt Kammer-Kerwick** [1,*] ◉**, Mayra Yundt-Pacheco** [1], **Nayan Vashisht** [1] ◉**, Kara Takasaki** [1] **and Noel Busch-Armendariz** [2] ◉

1 BBR, IC2 Institute, The University of Texas at Austin, 2815 San Gabriel Street, A0300, Austin, TX 78705, USA
2 IDVSA, Steve Hicks School of Social Work, The University of Texas at Austin, 1925 San Jacinto Blvd., Austin, TX 78712, USA
* Correspondence: mattkk@ic2.utexas.edu

**Abstract:** This paper describes a process that integrates behavioral and decision science methods to design and evaluate interventions to disrupt illicit behaviors. We developed this process by extending a framework used to study systems with uncertain outcomes, where only partial information is observable, and wherein there are multiple participating parties with competing goals. The extended framework that we propose builds from artefactual data collection, thematic analysis, and descriptive analysis, toward predictive modeling and agent-based modeling. We use agent-based modeling to characterize and predict interactions between system participants for the purpose of improving our understanding of interventional targets in a virtual environment before piloting them in the field. We apply our extended framework to an exploratory case study that examines the potential of worker centers as a venue for deploying interventions to address labor exploitation and human trafficking. This case study focuses on reducing wage theft, the most prevalent form of exploitation experienced by day laborers and applies the first three steps of the extended framework. Specifically, the case study makes a preliminary assessment of two types of social interventions designed to disrupt exploitative processes and improve the experiences of day laborers, namely: (1) advocates training day laborers about their workers' rights and options that they have for addressing wage theft and (2) media campaigns designed to disseminate similar educational messages about workers' rights and options to address wage theft through broadcast channels. Applying the extended framework to this case study of day laborers at a worker center demonstrates how digital technology could be used to monitor, evaluate, and support collaborations between worker center staff and day laborers. Ideally, these collaborations could be improved to mitigate the risks and costs of wage theft, build trust between worker center stakeholders, and address communication challenges between day laborers and employers, in the context of temporary work. Based on the application of the extended framework to this case study of worker center day laborers, we discuss how next steps in the research framework should prioritize understanding how and why employers make decisions to participate in wage theft and the potential for restorative justice and equity matching as a relationship model for employers and laborers in a well-being economy.

**Keywords:** labor exploitation; labor trafficking; intervention framework; agent-based models; disruption; collaboration; worker centers; well-being economy

## 1. Introduction

This study proposes the extension of a behavioral framework, introduced by Battista and colleagues [1]. This framework applies broadly to problems where data are scarce, the system is only partially available, and classes of ecosystem participants have conflicting objectives. We apply the extended framework to understand interventions in the problem domain of

labor exploitation and trafficking. The current study builds on research by the authors [2] that examined the potential of worker centers as a venue for introducing interventions that disrupt illicit behavior and promote cooperation between workers and employers.

For our study, the term day laborer refers to a person that is hired for a job, without the protection of a formal contract with an employer. Jobs are usually temporary positions that do not offer job security or additional benefits. These jobs are part of the informal sector, referring to the part of the economy that the government does not regulate with taxes or social welfare benefits. Without the protection of a legally enforceable contract, day laborers are vulnerable to a wide range of types and intensities of labor exploitation. This project adds to extant literature with similar goals in the exploitation of wildlife, community operations research, social and restorative justice, and the well-being economy [3–12]. A case study in labor exploitation and trafficking is used to demonstrate the utility of the approach.

We have organized the presentation of our research as follows. We introduce our approach to developing appropriate evidence-based interventions through complex sociological system analysis and modeling (Section 1.1). Then we provide background on our application domain of human trafficking and exploitation (Section 1.2) as behavior that can be addressed through our decision-science-based extension of [1] to iteratively develop interventions (Section 2). We then apply the proposed extension (Section 3) to a case study that focuses on wage theft as the most common form of exploitation experienced by day laborers [2]. It is important to note that, as an iterative process, our application of the proposed framework in this example case study involves the first 3 steps. Specifically, we utilize an agent-based model to examine the efficacy of educating day laborers about their rights and reporting options if they experience wage theft. Finally, we experiment with the model to hypothesize additional interventional targets. In Section 4, we continue our presentation with a discussion of future iterations needed for the development of effective interventions, following the extended framework. The paper concludes with a summary of the research presented.

### 1.1. Approach: Intervention Design for Complex Sociological Systems

An interventional framework can be analyzed through the lens of a complex sociological system that encompasses all relevant agents, interactions, and environments [8]. Past literature has defined a social system as 'complex' if ongoing interactions exist between active agents within some established organizational structure. These structures are self-organized by either the agents themselves or molded through pressure from external systems [13]. In either case, the organizational process is maintained through the system status quo, which all agents either adhere to passively or actively enforce through interactions [14]. If these interactions continue and become habitual, any intervention which seeks to change interactions by disrupting the system will be dismissed by at least one set of agents in favor of maintaining the status quo [15].

The stability of a complex sociological system depends on an adherence to the 'norm' —A pre-established understanding and expectation of how specific interactions will go [15]. These norms are foundational to the system operation and essential for agents' engagement with one another [15]. Within the complex sociological system of the informal labor sector, the most apparent interactions are between an employer agent and a worker agent. Those outside the informal sector maintain the expectation that (1) an employer hires a worker, (2) the worker does whatever labor was agreed to, and (3) the worker is paid, after which both agents part ways. This is the external status quo expected of a formal work sector. Previous research into the informal sector worker-employer interactions revealed that most day laborers had experienced some form of labor exploitation, the most common of which was wage theft [2]. A 2019 survey found that 86% of day laborers had experienced some form of exploitation, with 66% reporting wage theft. Although wage theft may not be a generally accepted status quo, it is a norm in specific informal sectors, thus part of the status quo of the sociological system [2]. Therefore, worker agents in this system make decisions based on the understanding that they may not be paid in full but

being paid in part is preferred to not being paid at all [2]. Interventions to change the outcome of such interactions have most commonly been implemented through a top-down approach [15]. While the problem and desired outcome are identified correctly, the success of past interventions has been lacking [7]. Agents must adopt an intervention as part of the system to see significant changes [8].

In the past, methodological advances to evaluate an intervention's development, practicality, effectiveness, and implementation were considered primarily theoretical, with limited evidence of in-practice application [15]. Guidance frameworks such as the 2008 UK's Medical Research Council were instrumental in reforming public health research design by assessing complex interventions and the systems in which they operate. However, the 'whole systems' approach has been criticized for overlooking critical aspects of the system itself [16]. The most common response is an intervention focusing exclusively on the victims. Obtaining information from victims to inform a response is the accepted model for intervention creation [15]. However, this approach only brings in one-half of the participants in the established sociological system. Community behavioral theory suggests that behavior change is unlikely when only one party is being directed to modify their response [15]. In the case of day laborers and exploitative employers, an intervention approach must address all participants in the system and community, rather than attempting to change the system itself or only a part of it [8].

Integrating an intervention into a community instead of blunt enforcement deviates from previous decision-making and behavioral science literature. Intervention frameworks such as the Battista 5-step process [1,17] focus on promoting system change through a change in the agents' behavior. Such an approach is rooted in a holistic understanding of why, how, and when the participants within a system operate. The interventions can then be implemented by drawing from both victims and perpetrators to achieve the desired outcome.

Gomes et al. (2018) discuss how community-based operational research can support the development of solutions to "complex, messy problems related to public goods" [7]. Research on such complex systems has benefited from the inclusion of qualitative methods, such as observation, interviews, and surveys among ecosystem participants, to expand the evaluative framework in an iterative, adaptive process to improve understanding of fit, adoption likelihood, use propensity, barriers to adoption, barriers to successful usage, and the likelihood of referral to others. For example, Arem et al. [18] report on a mixed-method operations research evaluation of a randomized trial for an antiretroviral therapy that assessed the efficacy of peer health workers in a low-resourced community in Uganda. This assessment incorporated in-depth interviews, focus groups, and direct observation sessions in addition to clinical data. Customizable analytic methodologies are needed for successful multidisciplinary effort(s) focused on specific social-ecological systems. In the current study, we apply agent-based modeling (ABM) to examine the employer-day laborer system dynamics.

### 1.2. Application: Labor Exploitation and Trafficking

Legal structures and law enforcement interventions to respond to exploitation are numerous and largely ineffective because they are reactive to events that are partially observable in informal processes that generally involve little documentation. We propose strategies informed by a public health perspective that focus on multiple levels of prevention, including primordial, primary, secondary, and tertiary. Further, we propose a broad decision-making framework that is fit for developing such interventions from early exploratory research, as examined in this manuscript, through to randomized control trials at scale.

Taylor [19] reviews the Palermo Agreement's 3Ps, adopted in 2000, and the Ruggie Principle, adopted in 2011. The former established a focus on prevention, prosecution, and protection for antitrafficking efforts. The latter established that governments and businesses have the duty to protect workers built from the 3Ps. Taylor observes that the limited impact of much of what has been implemented to address exploitative practices is due to a transactional focus rather than attempting to transform systems that contain those practices. The focus

shifts to extreme cases that meet specific legal requirements. Taylor concludes, "I argue that undermining the anti-trafficking cause to more directly challenge the systems producing everyday abuses within the global economy should be a goal, if not a moral imperative, for anyone serious about making workers' lives better" [19]. To these same ends, we encourage policy and social change efforts that focus on primordial and primary prevention. Primordial prevention [20] "focuses on the alteration of societal (i.e., environmental, economic, social, behavioral, cultural) structures that affect disease risk".

Past attempts to mitigate labor exploitation have focused on protecting victims and prosecuting offenders rather than attempting to fix the root causes of exploitation and the systems that allow such abuse to occur in the first place. A study of forced labor and human trafficking by the Issara Institute found that out of 81,690 workers in Cambodia, Myanmar, and Thailand, 19,978 met the international definition of forced laborers [19]. The most common exploitative behaviors endorsed in that report were being overworked, underpaid, deceived, coerced into recruitment, threatened, abused, and placed in debt bondage. Further, attempts at combatting instances of labor exploitation occasionally involved government intervention. However, remediation efforts most often came from the employer. Since day laborers are often self-employed and do not have the protection of a larger entity, the supply chain and retail partnerships that could provide protection for laborers are usually lacking. Taylor suggests that an inclusive approach is the most effective solution, explicitly developing intervention mechanisms that encourage, advocate for, and partner with the day laborers themselves [19].

Day laborers are especially at risk for exploitation since not all forms of wage theft result from the widely accepted definition of forced labor or human trafficking. This has allowed many day laborers in financially extractive (wage below the legal minimum) but not explicitly coercive (threats or physical violence) situations to slip between the cracks. A recent report in Australia found that over half of migrant workers were paid below the minimum wage [21]. Yet, the focus on interventions continues to be on workers suffering more outright, severe forms of exploitation.

Prior research has examined policies to remediate the problem of human trafficking, specifically labor exploitation, from the perspective of assessing the efficacy of existing laws and making recommendations for improved remediation [22–25]. Davy [26] systematically reviewed the human trafficking intervention literature, concluding that intervention evaluation is a substantial need. Davy found that although there are hundreds of anti-trafficking intervention programs and millions of dollars spent annually, the impact of these programs is relatively unknown due to limited program evaluation. Improving the quality and frequency of evaluations is critical [26]. Data collection to inform future anti-trafficking interventions has failed to include victims in the data collection process, thus reducing the quality of the data relating to the program impact, the extent of trafficking, and how victims think the situation could be improved [26].

The reasons why employers exploit their employees, especially day laborers, have not been studied extensively due to the reluctance of employers to admit their illegal practices. Past literature on systemic reviews of factors that contribute to exploitation has identified that: acceptance of wage theft [27], benefits [28], and misinformation [21] are commonly cited as primary reasons that wage theft occurs. Using these drivers as a starting point, our proposed extension of Battista's framework can be applied to develop a behavioral intervention.

## 2. A Framework to Develop Interventions Targeting Human Behavior

Battista and colleagues [1] proposed a behavioral science-based process to develop nominal interventions for illegal fishing in fishery systems with resource levels that constrain the use of strict sanctions and extensive monitoring. Instead, their process focuses on changing participant behavior within the system. This process focuses on the social and psychological factors influencing behavior, such as norms, expectations, trust, and perceived legitimacy of regulations as the foundation for interventions. By altering the motivation and behavior of agents, interventions are longer lasting and farther reaching than limited enforcement

capacities. The agents themselves enforce the social discouragement of undesired behavior [29]—Illegal fishing for Battista or wage theft in our study—Rather than relying solely on pressure from external forces. Rational agents will perpetuate illicit behaviors which fall within acceptable norms, expectations, and beliefs [30]. By introducing interventions that alter these factors, continuing to act in an undesirable manner becomes too costly, and the behavior begins to change. However, people are not always rational actors [31]. Often their actions are due to automatic processes and mental unawareness [32]. Battista's framework targets for intervention the factors that most strongly influence illegal behavior.

The Battista process has broad application and is adapted here based on the lessons from a case study of labor exploitation and trafficking of day laborers. Their process focuses on changing specific illegal fishing behaviors and begins with characterizing beliefs, norms, ways of thinking, and ways of acting relative to illegal fishing. The process continues with artefactual experiments to pilot potential interventions before implementing them at scale. Specifically, Battista and colleagues develop a 5-step process, included below and in Figure 1 as Steps 1, 2, 4, 6, and 8. Motivated by our case study, we propose additional steps 3, 5, and 7 to illustrate where community operations research, specifically through simulation and machine learning, can be incorporated into this process in a manner that integrates behavioral and decision science to improve understanding of the problem domain.

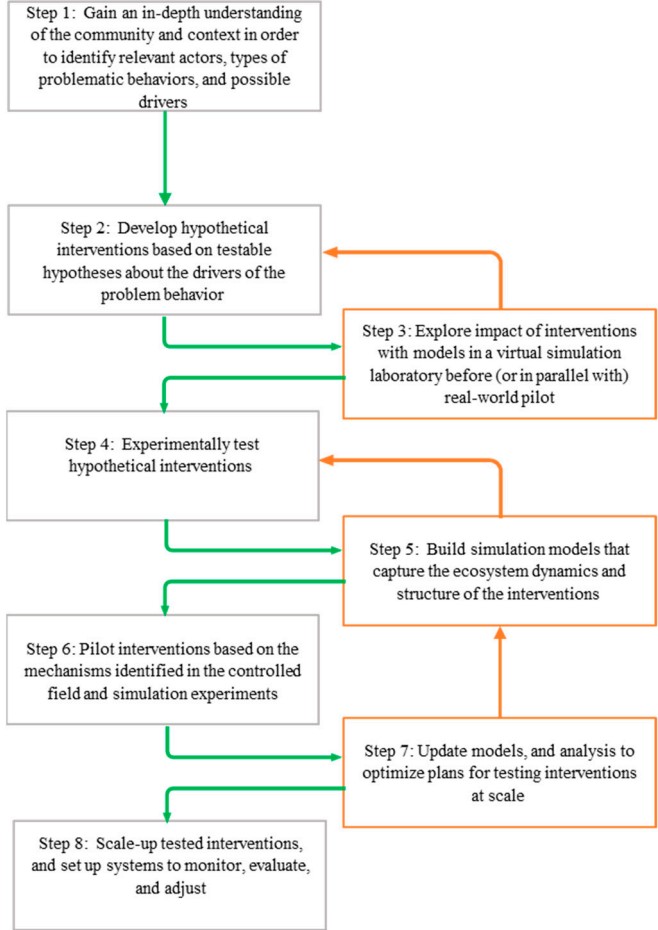

**Figure 1.** Behavioral and Decision Science Framework to Design and Implement Interventions. Figure 1 displays the proposed behavioral and decision science framework. The original five steps (1, 2, 4, 6, 8) have been augmented with steps 3, 5, and 7, wherein modeling and simulation are used to improve the design and implementation process.

### 2.1. Extending the Behavioral Framework by Battista and Colleagues

Starting with the 5-step Battista process for implementing the intervention, the ABM supplementation adds three more steps (3, 5, and 7) to the process. Step 1 aims to gain an

in-depth understanding of the community and context to identify relevant actors, types of problematic behaviors, and possible drivers. Step 2 is to develop hypothetical interventions based on testable hypotheses about the drivers of the problem behavior. The theories and hypotheses should formalize the relation between the expected behavior, including contextual factors and the individual circumstances of those displaying the problem behavior. Step 3, the first instance of the ABM -enhanced process, is to build simulation models that capture the ecosystem dynamics and structure of the interventions. Research and data gathering as part of ABM development will enhance the learning from Steps 1 and 2. Step 4 is to experimentally test hypothetical interventions on populations resembling the target population, using artefactual field experiments and additional simulated experiments to test hypotheses developed in Step 2. Step 5, the second ABM addition, explores the impact of interventions with models in a virtual simulation laboratory before (or in parallel with) a real-world pilot. Step 6 pilots interventions based on the mechanisms identified in the controlled field and simulation experiments. Step 7, the final ABM step addition, is to update models and analysis to optimize plans for testing interventions at scale. The final step, Step 8, is to scale-up tested interventions and set up systems to monitor, evaluate, and adjust to fit different contexts for retesting.

The benefits of this integration follow a theme: study the system toward the goal of developing and testing interventions using participatory methods and simulation and modeling methods to learn more about the system virtually to reduce risk and time when testing or deploying those interventions in the real-world system. The three added decision science steps in the behavioral science process utilize essential artefactual data from local communities as ground truth to enhance modeling and analysis. They also incorporate empirically supported utility functions for the decision-making of ecosystem participants in a manner that allows multiple criteria to be considered, thereby reducing the likelihood of experiencing unintended consequences in deployment. The resulting integration provides a framework for generating and collecting evaluative data that can increase the effective deployment speed and efficacy of deployed interventions. The case study that is integral to this project incorporates the first three steps and informs our plans for future research that corresponds to an iteration back through Steps 1 through 3 before incorporating additional experiments as we plan for Steps 4 through 8.

### 2.2. Utility of Simulation and Agent-Based Models in the Extended Framework

Our additional Steps 3, 5, and 7 incorporate ABMs, which we introduce before presenting our case study. Macal [33] provides an accessible tutorial on ABMs for readers who are new to the approach. As part of this introduction, we connect several reviewed models with the model we developed for this case study.

We begin this introduction by citing observations made by Lindkvist [34], who examined the utility of ABMs in the problem domain studied by Battista and colleagues: the sustainable governance and management of small-scale fisheries. Lindkvist [34] identifies three main challenges of small fisheries management that are addressable by ABMs: (1) improving the way collective action and heterogeneity in human behavior can be incorporated into research and management, (2) developing policies that are sensitive to local contexts while also accounting for regional and global contexts; and, aligning with Battista and colleagues, (3) tackling data scarcity and uncertainty. While all three challenges are relevant to our framework and case study, data paucity is particularly salient. As commented above, complex systems are often partially observable, and ABMs provide a means of integrating disparate sources of qualitative and quantitative data into a structurally realistic model that can generate synthetic data about the system. Some caution is warranted, however: model development can be slow due to the need to study complex processes, and balancing the ability to characterize complexity while retaining interpretability can be challenging [34]. In wildlife and ecology management, broadly, ABMs are "capable of simultaneously distinguishing animal densities from habitat quality, can explicitly represent the environment and its dynamism can accommodate spatial patterns of inter- and intra-species mechanisms,

and can explore feedbacks and adaptations inherent in these systems" [35]. Agent-based simulations reveal emergent behavior in more complex real-world modeling environments and generate data that can be used in other models and analyses that, in turn, are used to plan and deploy interventions to meet policy objectives. Constructing an ABM is valuable as part of observing the consequences of policy and social theory in an artificial environment based on reality. ABMs can trace the observed interactions among agents back to actual individuals while providing feedback on each agent [36]. While constructing the model, researchers are tasked with establishing what decision-makers need to know about specific parameters to fully understand the potential for the behavior being studied [36]. In the case of our model, that behavior is whether a laborer will accept a job, and the model allows that behavior to emerge under various circumstances that include opportunities for the worker as well as exposure to hazards of exploitive employment.

Utomo [37] the literature on using ABMs in agricultural food supply chains. Their review reveals that most ABM studies have focused on licit aspects of agricultural food supply, including production planning, investment, technology adoption, cooperation and partnerships, product quality, selling, and distribution. They develop a research segmentation and discuss gaps in the literature that can be addressed with future research. Notably absent from the studies reviewed and their discussion is studies dealing with worker rights and exploitation, the subject of our case study.

As a notable exception to this pattern, and a motivating example to the current study, Chesney et al. [38], used an ABM approach to study labor exploitation in the Spanish agricultural sector, confirming that various socioeconomic aspects of labor supply and demand increase the likelihood and degree of exploitation, including labor trafficking. Their study, in turn, builds on a framework from Crane [29] that characterizes labor exploitation and trafficking as a management practice that includes five enabling conditions for such exploitation: industry context, disadvantaged populations, geographic context, cultural context, and regulatory context, all of which have relevance in the case study for this project. Chesney et al. [38] use an ABM to investigate the propositions developed by Crane [29], implementing employer and worker agent types. In summary, employers aim to hire workers at a minimum cost. Workers can accept or decline a job offer and have the option of leaving an area if they can't find work or earn enough. Employers can change the amount they pay workers unilaterally as a percentage of the promised amount. The ABM generated experimental data using graphical and regression methods [38].

Zhang et al. [39] developed an ABM to study the safety behaviors of construction workers and how they interact with managerial safety policies. They view "safety performance as an emergent property of the behaviors and interactions of construction workers and management teams". The behaviors of the agents in their ABM were informed by two surveys conducted across different classes of workers, including managers and safety professionals, and construction workers. The research process to develop the model included visits to construction sites to observe and record unsafe behaviors. Ultimately, their model included agent classes for workers, supervisors, safety officers, and senior management.

In the ABM constructed by Busby [36], media is used as the central risk communication intervention, which connects risk responses with decisions and behaviors of agents within the model. The interactions that occur in natural disaster models are reflected in how the worldview of agents affects their tendency to amplify risk, more specifically, how certain actors or interventions can be used to influence opinions, risk assessment, and future decisions [36]. The information diffusion for modifying behavior occurs when the social actors find out about risks from other actors, and so update their own beliefs [36]. The model demonstrated that the risk perception of individual actors correlates with and influences people like them most effectively. Busby and colleagues included risk principles, which are social actors that fail to act appropriately and whose reputation affected the action of agents who interacted with them [36]. In the case of natural disasters, these risk principles were governmental organizations; for our case study, the risk principles which day laborers interacted with include the Texas Workforce Commission, OSHA, and

exploitative employers. Finally, Busby and colleagues introduced risk communications, which are influenced by the chosen risk communication intervention, such as the news, media, or public service announcements [36]. In our case study, we incorporate a media campaign as an intervention to augment the effect of direct action by advocates.

## 3. Extended Framework Application: Day Laborers & Exploitation Interventions Study

Authors [2] report on a case study that included three data collection phases of in-depth interviews among day laborers in the Houston area in 2016 (Study 1) and again in 2018 (Study 2) and a set of two-part interviews in the Austin area in 2021 (Study 3). The current study builds directly from this prior work [2,40], and we include a summary here for continuity and to connect it to the proposed framework.

Our three-part case study builds on prior empirical evidence about day laborers in Texas, partly motivated by examining how natural disasters might exacerbate exploitive dynamics for precarious workers in construction.

### 3.1. Background–Artifactual Study of Day Labor in Texas

In the immediate aftermath of Hurricane Harvey, Theodore [41,42] surveyed the Houston area's day laborers (n = 361). Data were collected at 20 informal hiring sites located in Houston and Pasadena. Post-disaster, immediate risks faced by workers—Such as injuries, infection, and rushed hiring of crews–spiked, with 64% of undocumented workers stating that they would not seek help for emergencies or report violations to government agencies out of fear of deportation [41]. Among the findings, Table 1 summarizes those results pertinent to the current project [41]. More than half of the day laborers in the Theodore study had experienced wage theft in the first month after Hurricane Harvey. They found work on about 2.5 days per week and had low awareness of the agency responsible for addressing wage theft violations, in this case, the Texas Workforce Commission. They also had low levels of familiarity with organizations that might be able to assist them in recovering stolen wages. Last, they were often asked to perform tasks beyond those they were hired to perform, and few had received any training. This study concluded that worker centers are critical disaster recovery hubs [41] during reconstruction clean-up efforts.

**Table 1.** Summary of Results from Theodore (2017).

| Summary Metrics for Day Laborers (n = 361) | Result |
| --- | --- |
| Hour wage paid | $13.40 |
| Median hour per day worked | 8 |
| Median daily wage paid | $100 |
| Days of work per week | 2.5 |
| Experienced wage theft | 57% |
| The average amount of wage theft per instance | $225 |
| Percent that was asked to perform tasks beyond what they were hired to perform | 61% |
| Percent that was aware of the agency responsible for wage theft violations | 0% |
| Percent unable to name an organization that could provide wage theft assistance | 92% |
| Percent who had not received any training for the tasks they were hired to perform | 85% |

### 3.2. Summary of Prior Three-Part Case Study

Study 1 [2,40] included 44 interviews (22 men and 22 women) at street corners, provided rich qualitative information about the lived experience of day laborers, and quantified the rate at which they endured a range of exploitive behavior, including abusive labor practices and human trafficking violations. This survey revealed the following structural aspects of decision-making by day laborers:

- Day laborers generally have imperfect information about the job that they are offered and decide about that job myopically or choose to wait for another job opportunity.
- Occasionally laborers employ risk mitigation strategies, but economic pressures are usually sufficient to induce accepting a job offer with imperfect information. These occasional strategies include:
  - Safety in numbers (a group of laborers seeks employment together)
  - Waiting for a trusted employer (employer reputation is a critical factor, and to the degree that they can, laborers prefer employers that are known to them.
- Often the laborer does not know their actual employment state until the job or day is finished. The employer may pay them in full, partly, or (occasionally) not at all.
- Full pay is typically $120 per day but can range between $80 and $150.
- Jobs frequently last multiple days (1–4 days is typical)
- Wage theft occurs between 10% and 30% of the time.
- Wage theft results in the loss of between 10% and 25% of income earned.

Study 2 [2,43] included 19 interviews (17 men and 2 women) with day laborers at street corners and examined the decision-making processes by day laborers when seeking work, including the trade-offs they are forced to make when navigating the hazards present in their precarious work environment. This study revealed that the employer's reputation for paying the worker as agreed and for providing a safe work site substantially impacts participants' decisions to choose a job. Similarly, a worker's likelihood of a job being accepted when the safety condition is perceived as entirely safe is substantially higher than when the site has little to no safety precautions. While these results were based on a small number of interviews, the model provided a means of characterizing the importance of worker perceptions about the employer and the job site, with implications on benefits to workers of having more reliable information about employers.

Study 3 [2] included 36 interviews with male day laborers contacted through a worker center and investigated the potential of providing day laborers with training designed to increase their knowledge of their rights as workers and the options that they have if they experience labor exploitation and trafficking. This study included two interviews: the first provided participants with an experimental manipulation wherein they were read a statement about their rights as workers and allowed to discuss and ask questions about this material. The first interview measured their likelihood of reporting a future event of wage theft before and after the experimental manipulation. They were also asked about the likelihood of sharing that information with other workers. The second interview, completed by 28 of the original participants, was conducted 48 to 72 h later and covered their recall of the information and the likelihood of sharing it. Specifically, although this was a small experiment, these artefactual findings suggest that education among those who have experienced wage theft has the potential to increase their likelihood of informal reporting if they have never reported before. Of relevance to the current study, we observed a 50% increase in the likelihood to report wage theft after a single dose of education about worker rights and reporting options among those workers who had experienced wage theft but had not reported it.

### 3.3. Implications for the Current Study

Through these artifactual experiments, we observed that, regardless of whether a worker discusses the pay, site conditions, specific job tasks, or hours of work for a job, the reality of what the worker experiences on-site can vary significantly from their initial perception. The change from a worker's expectation to experience is due to actions by the employer, who regulates what occurs from the moment the job begins until after it has ended. Within the informal day laborer sector, and due to personal circumstances, a worker may accept a job they assume will be unfair, whether in pay theft, extended hours, unsuitable work conditions, or numerous other violations that constitute worker exploitation and labor trafficking. The point when a worker must decide whether to accept or forego a job offer made by an employer is when the worker holds the most agency

in the laborer-employer interaction. Throughout the job and immediately afterward, the employer holds the most power. The worker can regain that power and agency should they decide to pursue justice and disclose instances of exploitation through nongovernmental agencies, advocates, and other legal avenues.

Training for worker center employers, paired with incentives to implement fair employment practices [44], can curb repeated wage theft more effectively than employer training alone. Primary prevention programs, law enforcement support, and continued education for workers and employers are different avenues that all work to reduce abuse [45]. We have summarized these dynamics and intervention targets in Figure 2.

Artifactual Study of Laborer Employment: Journey Map with Interventional Targets for a Worker Center

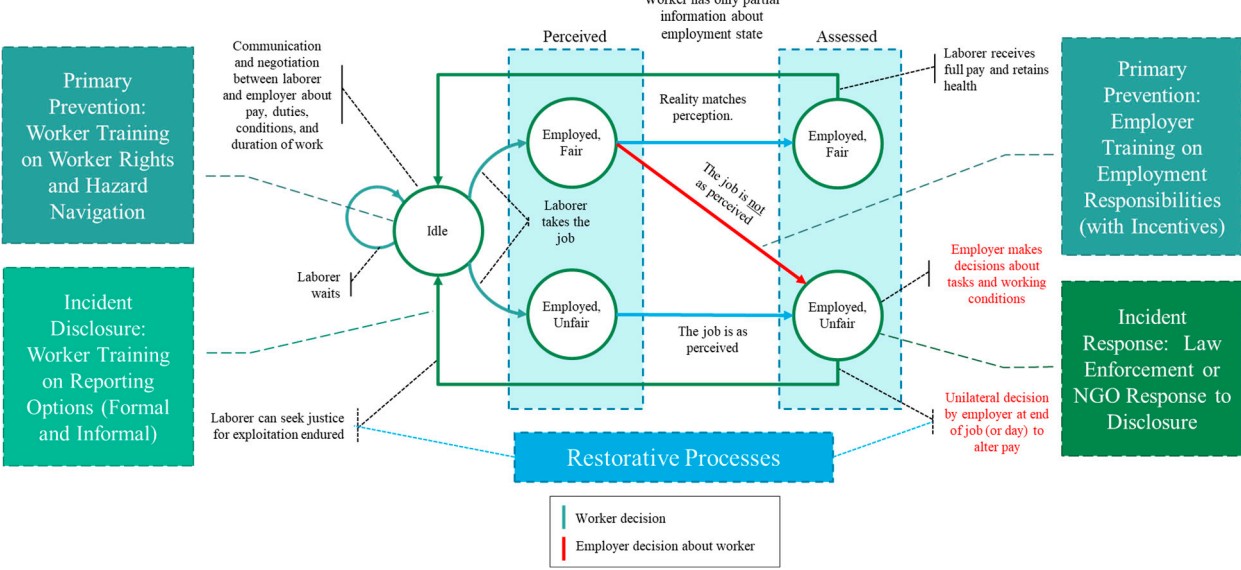

**Figure 2.** Interventions and Interventional Targets to Ameliorate Poor Labor Conditions. Figure 2 displays a typical journey cycle for day laborers as they navigate decisions about opportunities for work and some of the hazards in that employment ecosystem. Select interventional targets are shown, including primary prevention options that might be integrated into a worker center and coordinated law enforcement interventions [2].

Interventions aimed directly at the workers– education, talks, training, advocates, pamphlets, etc.—Are most common [46–48]. The success of such interventions varies and depends primarily on how integrated the center is with the day laborer community [47]. Worker centers that community members operate here to the culture and language of the day worker demographic are in areas with a high likelihood of worker-employer interactions and have high engagement rates [49,50]. Additionally, community-centered workers' centers provide a clear and open line of communication between center employees and day laborers. As trust builds, day laborers feel secure in sharing concerns and instances of abuse with employees, who can introduce or adjust interventions or aid in addressing the exploitation [50]. The effect of interventions becomes more effective as the target population becomes increasingly enthusiastic about the interventions [51].

### 3.4. The Agent Based Model

The current study draws from our artifactual study of day laborers (Steps 1 and 2) to develop an ABM sufficient to examine various interventional candidates in a virtual environment (Step 3) before conducting pilot studies (Step 4) or RCTs at-scale in the field (Step 6). Figure 3 displays the translation of the day laborer journey map of Figure 2 into a multi-agent, state, and decision diagram that we implemented as an ABM. In our model, there are agent classes for (1) laborers, (2) employers, and (3) interventions. While all three are dynamic objects created in the model that are endowed with data properties and computational functionality,

the first two are the human participants in our system. The intervention agent could be a human (advocate) or control policies (PSA) to be added to the model.

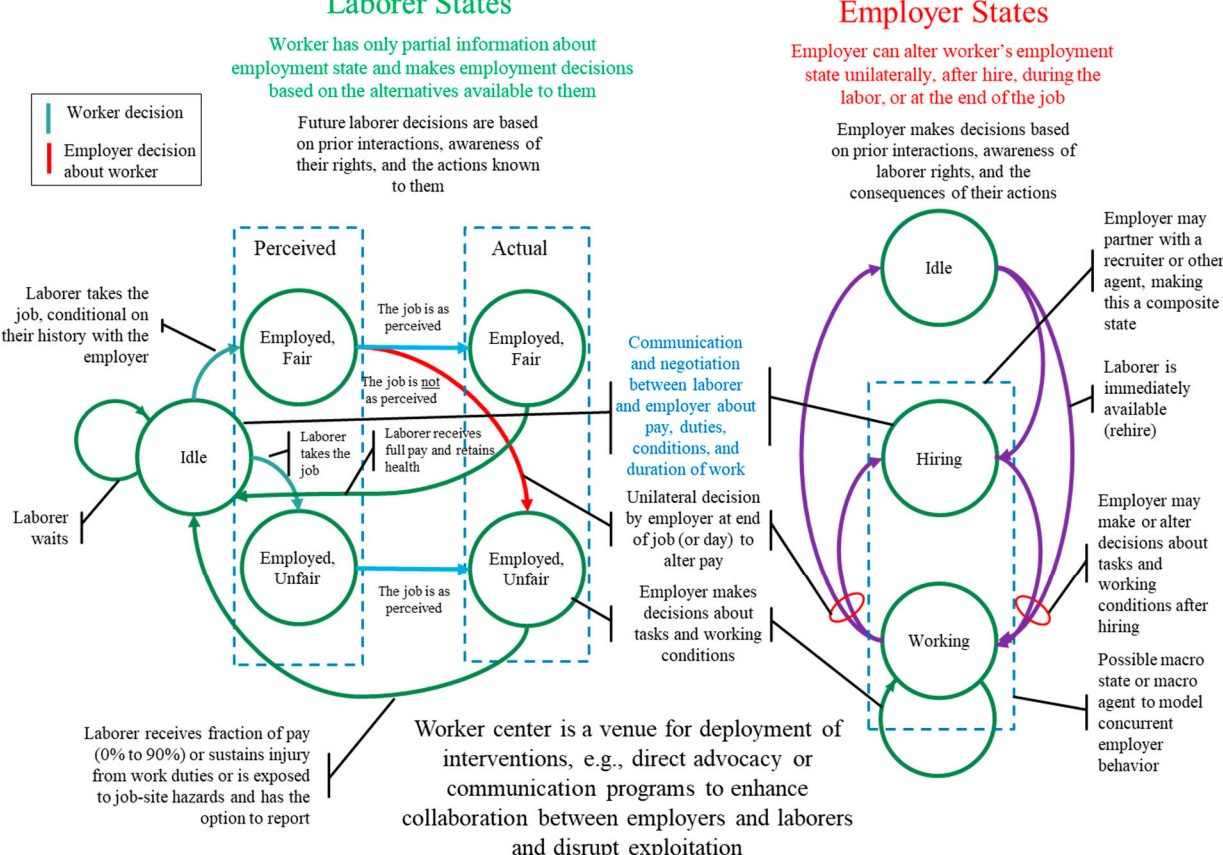

**Figure 3.** Relationship between Labor and Employer State & Decision Spaces. Figure 3 displays the decision-making process associated with the day laborer journey of Figure 2, now identifying the agent classes, the states for those agent classes, and the possible transitions, both controlled and uncontrolled, that are incorporated in the ABM developed from the case study.

In this translation, we identified the relevant agent classes for participants in the ecosystem (laborers and employers), the states for those agent classes, and the transitions that are possible, both those that can be controlled by actions taken by an agent and those that occur without control, either associated with a stochastic event or a deterministic outcome defined by dynamics modeled in the ecosystem.

Further, we incorporated the training manipulation examined in the case study as a third category of agent implemented as a type of Bass diffusion model (BDM). Within the BDM approach, we implemented a modality to include the effect of advocates working directly with laborers in the field and a modality associated with a public awareness campaign (PSA). We introduce the BDM and discuss its use in similar systems and our ABM next.

### 3.5. Bass Diffusion within ABM

The BDM captures the dynamics of the response and rationale of a population when presented with a new product and documents the adoption process of the product [52]. For our study, the 'products' introduced to the labor agents are anti-exploitation interventions: advocates and public service campaigns. The product adoption rate throughout the population in a BDM depends on exposure to the product and feedback from adopters to non-adopters [52].

The dynamics of the model are defined by a differential Equation (1) which describes the process of a new product, information, or diffusion into a population [52].

$$\frac{dN(t)}{dt} = (m - N(t))\left[p + \frac{q}{m}N(t)\right], \ t \geq 0 \tag{1}$$

where

$N(t)$ ≡ number of people converted at time t as part of the campaign.

$m$ ≡ size of population

$p$ ≡ conversion as a direct consequence of the campaign (innovation)

$q$ ≡ conversion from word of mouth

The degree of innovation, $p$, represents the external motivation for adoption, while the degree of imitation, $q$, is the effect of word of mouth in the diffusion of a product. In the present case study, $p$ and $q$ are coefficients that were determined by reviewing previous research. The quantity $m–N(t)$ is the number of members of the population who have not adopted at time $t$. The quantity $p + q*(N(t)/m)$ is the probability of adoption at time t, and incorporates the direct effect of the campaign, $p$, and word of mouth, $q*N(t)/m$, from those who have adopted.

Bastani [45] used a BDM within an ABM to simulate the diffusion of energy-saving policies among the occupants and the related impact on energy consumption of commercial buildings. The agents in Bastani's BDM have a rate of contact (ROC) in which one agent can contact several agents to introduce energy-saving policies [45]. The rate of contact illustrates the number of connections each agent will attempt to make and is related to word-of-mouth (WOM) diffusion adoption. Multiple trials of the Bastani model revealed that the effect of word of mouth among occupants had the strongest influence in persuading occupants to save energy, the other alternative manner of information diffusion communication included media [45].

In the current study, the ABM was developed to allow for two forms of intervention to be tested: direct training by advocates alone and supplemental media campaigns in addition to direct advocacy. This modeling decision was based on strategies we have observed by service providers and communities who add media campaigns to advocacy based on the availability of resources. Our case study's educational manipulation informed how direct training was incorporated into the model.

The specific motivation for supplementing direct advocacy with the media campaign is drawn from a campaign developed by the City of Houston utilized in the reconstruction period after Hurricane Harvey [53] and interviews with advocates. The campaign, Build Better Houston, had components that were motivated by post-Harvey reconstruction efforts and policies adopted in the rebuilding efforts in New York after Hurricane Sandy. Among other protections, the program included a $15 per hour base wage, workers' compensation insurance, and Occupational Safety and Health Administration (OSHA) construction certification training. Other initiatives, including executive orders for city procurement, operational policies, and media campaigns, are documented in the City of Houston's Anti-Human Trafficking strategic plan. At a high level, this plan establishes a complementary community-based paradigm that adds a nontraditional municipal response and public health approach to a traditional law enforcement approach.

This campaign was not implemented with any measurement components but was executed through several media channels. Thus, the dynamics of the media campaigns used in the model are based on extant literature about how information diffuses through a community. Specifically, the model incorporates dynamics from the BDM.

### 3.6. ABM Specification and Development

The following sections describe key programming aspects of the ABM which were designed to incorporate the relationships in Figure 3 and the influence of the planned interventions. The model was programmed in AnyLogic and included custom Java implementation of many sub-models guiding an agent's behavior, as discussed in detail

below. As a novel feature, Laborer agent behavior includes memory decay and access. It is important to note that while many aspects of the ABM are based on observed behavior in our empirical exercises, some aspects of this demonstration model have been specified in ways that we hypothesize the system behaves. This aspect of the model design is consistent with the iterative process in the intervention framework, and we propose future research to further study these hypothesized dynamics. Prominent among our proposed future research is a deeper examination of the decision-making processes of employers.

*Agents:* In addition to agents for laborers and employers, described earlier, the ABM implements two different intervention agents that model high-level diffusion through an advertising campaign and a more direct "word-of-mouth" approach, as well as other features relevant to this scenario, such as a laborer reporting feature when a laborer has faced exploitation. More information about the ABM and our implementation in AnyLogic is available as a supplementary document from the corresponding author.

The model is initiated with a set of laborer and employer agents. The interaction between these agent classes is initiated, as shown in Algorithm 1. An employer agent creates a job and announces that job to a nearby available laborer agent. The job is an abstraction within the ABM that enables dynamic behavior between laborer and employer agents and consolidates variables and methods that are used/accessed extensively by them. The laborer agents are endowed with the ability to form and store memories about past employment experiences.

---

**Algorithm 1**: Job Creation and Acceptance Decision Logic

---

**While** (Simulation Runs)
Employer agent creates Job(current_simulation_time)
Employer agent sends Job to single randomly chosen nearby Laborer agent
Chosen_Laborer.current_job = Employer Sent Job
**For** each memory in chosen Laborer agent
    **If** memory.Employer = = current_job.Employer:
        **If** memory.access( ):
            **If** memory.tolerate( ):
                Chosen_Laborer.accept_job = true
                **break**
            **Else**:
                Chosen_Laborer.accept_job = false
                **break**
**If** Chosen_Laborer.accept_job:
    Laborer and Employer leave Idle States and enter Working States

---

Mirroring our observations in the real system, the employer makes a decision about stealing the laborer's wage at the time the job is over and it is time to pay and move into the 'Paid' state. Literature and past interviews have revealed that the more dire the employer's economic situation is, the more likely they are to steal; on the other side, the more dire the economic situation is of the Laborer, the more likely they are to report. Technically, a job is created with the capacity to store a 'fairness' calculation. The value from this calculation determines whether the job will be fair—the actual amount paid out to the worker matches the perceived pay amount- and this calculation is only revealed after the job is complete. The agents' response depends on the socioeconomic characteristics drawn from literature and interview data.

At the same time, laborer agents—who begin in an Idle state as well—receive the said message and decide to decline or accept the job. We have assumed an exponential decay for this decision based on qualitative insights from our empirical research. As shown in Algorithm 1, if the laborer has the memory of a prior theft by the employer, they are likely to decline that job. However, it won't be long before they would accept a job again from the same employer. If no Laborer accepts the job, or if the job is declined, the job will eventually

time out, and the Employer agent will remain in an Idle State. If the Laborer decides to accept the job, both the Employer and Laborer move into a Working state.

Algorithms 2 and 3 illustrate the logic implemented in the ABM as pseudocode for how a laborer agent responds to wage theft based on whether they have received education about their rights and how they retain memories of the behaviors they have endured before re-entering the Idle state.

---

**Algorithm 2**: Job Completion and Reporting

---

**Wait** Chosen_Laborer.current_job.hours
    //Employer theft propensity functionality
    Employer pays Laborer Agent full pay or commits wage exploitation
    **If** Laborer.informed:
        Laborer reports Employer
        //Reporting functionality
Laborer creates JobMemory(current_simulation_time, current_job)
Laborer.memories.add(new JobMemory)
Laborer and Employer return to Idle States

---

In the ABM, the employer and the labor agents focus on the job that connects them. For simplicity in the current ABM, the employer can only focus on one job at a time. Once the job has been completed, the laborer agent receives a message from the employer agent, revealing if the job they were working was 'fair' or 'unfair'. If the job is fair, the Laborer will be paid and return to the Idle state until a new job message is received. An unfair job is characterized by the Laborer experiencing a certain amount of wage theft randomly generated from a uniform distribution whose bounds are set as parameters. At this point, the laborer agent is at a decision-making node in the ABM where they can choose to report a job if it is revealed to be 'unfair.' Based on our qualitative insight from empirical research, we have implemented the decision logic for reporting by a laborer agent as outlined in Algorithm 3. In the current reality, reports of wage theft are extremely rare. Therefore, we have devised logic that approximates this situation.

---

**Algorithm 3:** Reporting functionality

---

**If** Laborer.Cumulative_income is negative
    Report = true
**Else if** exposure to educational messages
    Report = true
**Else**
    Report = false

---

The calculation for reporting likelihood is implemented and activated in an unfair job. The reporting functionality is a cascading logic and is based on artifactual evidence. An informed laborer will report if (1) the theft results in the daily pay being less than their daily expenses and (2) the efficacy of the interventions. Laborers' likelihood of reporting is implemented as a function of educational messages. We have assumed an exponential relationship between the doses of education received and the likelihood of reporting. We have calibrated this curve (see Figure 4) to the results of an experimental manipulation by the authors. This manipulation involved a pre/post-education measurement of familiarity with reporting options. Specifically, the reporting likelihood prior to education is approximately zero and increased to approximately 0.5 in a survey of day laborers [2]. More training means there is a greater chance of reporting.

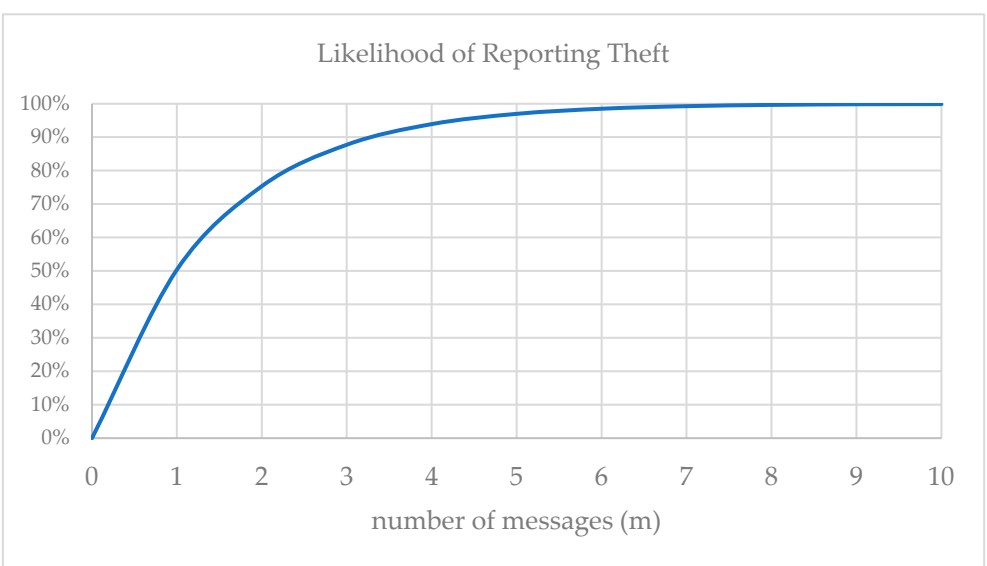

**Figure 4.** Learning Curve as a function of Number of Exposures to Education. Figure 4 displays the assumed learning curve as a function of number of exposures to education. Caption: This exponential relationship was calibrated to the results of an experimental manipulation by the authors. Specifically, the reporting likelihood prior to education is approximately zero and increased to approximately 0.5 in a survey of day laborers [2].

Laborers become informed about their rights and options as they receive repeated messages (from advocates, media campaigns, or other workers). A report made by the laborer affects the employer's future behavior. As their inclination to report goes up, the penalty for the employer goes up, and hence the overall wage theft goes down. This feedback loop is governed by the employer's theft propensity variable.

*Employer Theft Propensity:* We assume based on qualitative insight from our empirical research that day laborers' specific actions like reporting wage theft will reduce an employer's future propensity for theft. Specifically, in the ABM, reporting wage theft incurs a penalty for the employer. Once the employer is reported, the propensity for future theft is lowered. As more reports occur, the employer is less likely to commit wage theft, eventually tending to zero. The degree of decreased theft propensity depends on the effectiveness of reporting and the associated penalty. The likelihood of the reporting success will determine if the employer returns the stolen wages to the laborer. Additionally, if a successful report also includes punitive damages, then the future theft propensity on the side of the employer is calculated through the total funds subtracted by the sum of total stolen wages and an additional penalty, divided by the total funds. The resulting amount is how much 'punishment' the employer will endure if wage theft is reported. In the current ABM, future theft propensity for an employer agent *e* is adjusted through a scaling factor, $\Theta_{e,p}$ that is in the range 0 to 1, with a dynamic value set based on a report of wage theft by a laborer. Specifically, the propensity for theft is multiplied by $\Theta_{e,p}$, and that propensity is guaranteed to be less than 1. If there is a low chance of being penalized, the potential gain of exploiting workers is worth the risk. However, if an employer is penalized once, there is a deterrent in place to caution them against exploiting their workers again in the future. Although future research is needed to refine our understanding of how reporting impacts employer decision-making, these relationships have been operationalized in the model as follows, where *e* indexes employers and *t'* is future time. Each employer agent maintains a value of the total funds they have accrued as the proceeds of jobs they have created, Funds$_{\text{total}}$. We adjust $\Theta_{e,p}$ by subtracting stolen wages that are returned and any associated penalties from the total funds available to the employer. $\Theta_{e,p}$ is not allowed to become negative.

$$Propensity_{e,t'} = \theta_{e,p} * Propensity_{e,t}$$

If a laborer agent endures wage theft from employer *e* and decides to report:

$$\theta_{e,p} = \frac{Funds_{total} - (Wages_{stolen} + penalty)}{Funds_{total}} \; \forall \; \theta_{e,p} \in \; [0,1]$$

***Interventions:*** We incorporated the training manipulation examined in the case study as a third category of agent implemented as a type of Bass diffusion model (BDM) described above. Within the BDM approach, we modeled a modality to include the effect of advocates working directly with laborers in the field and a modality associated with a public awareness campaign (PSA). The ABM model has multiple intervention mechanisms that operate as separate agents. The primary avenues are Direct Intervention through Word of Mouth (Advocates) and Broadcast Intervention (Public Service Announcements). All interventions serve to transform idle or uninformed labor into an informed agent. Informed agents can spread information to other workers via word of mouth, thus creating more informed agents. As more worker agents become informed, their likelihood of reporting increases. Both Direct and Broadcasting Interventions have two main variables: contact_rate and adoption_rate. The Broadcasting Intervention has an additional variable: campaign_effectiveness. The interventions can occur at any point in the model for the worker—before, during, or after a job. Direct Interventions work through Direct Campaigns and Broadcast Campaigns. Word of Mouth Interventions works as a Direct Word of Mouth and Word of Mouth Broadcast. Most Word-of-Mouth work occurs through advocates or through workers who, after interacting with one of the interventions, become informed and speak to other laborers, who also become informed. The worker's inhibitions must be considered for the diffusion of Word of Mouth among the worker population. The conversion rate is associated with either direct or broadcast campaigns; the effectiveness of the informed worker's word-of-mouth information diffusion is a function of how they were informed. Campaign effectiveness is not utilized by the direct intervention agent as they only send out messages within their social distance network reach. Contact rate and adoption fraction are incorporated into the model. Reinforcement capability is put into the model as messages of enforcement from multiple channels are received.

*3.7. Model Validation*

The present study outlines the development of an Agent-Based Model (ABM) that incorporates the agents' interrelationships as depicted in Figure 3, as well as the effects of planned interventions. To ensure the credibility, precision, and dependability of the model and its outcomes, it is imperative to validate the ABM. Numerous validation techniques are described in the literature [54], and in this study, we adopted the framework proposed by [55], which has also been employed in a similar study by [56]. Specifically, we concentrated on the processes entailed in the structural validation technique that are pertinent to our ABM. The structural validation technique entails the generation of observed system behavior and encompasses the following processes:

- *Calibration*: The initial step in the validation process is calibration, which involves determining the model parameters using empirical data from the real world. This step is essential and should be conducted prior to model validation. In the present study, the parameter values (or ranges) used in our ABM, as presented in Table 2, were established based on insights gained from our own empirical research as well as from pertinent extant literature.
- *Sensitivity Analysis*: A crucial step in model validation is sensitivity analysis, which involves altering parameters and observing how the results change. This process helps to assess the model's robustness to changes in the parameters and identify the parameters that have the greatest impact on the results. In this study, the ABM was designed to test two forms of intervention: direct training by advocates and media campaigns. The sensitivity analysis of the model examines the behavior change of laborer agents due to the addition of a direct intervention and a media campaign that complement the effects of two advocates working with the day labor community. This

enables the impact of the Public Service Announcement (PSA) to be observed in a virtual environment, dependent on the effectiveness of the campaign. It is expected that these interventions will increase the likelihood of laborers reporting wage theft, thereby reducing the employer agents' propensity to steal and overall wage theft. Three different scenarios were tested in this analysis, each run for a two-year period with 30 replications of the simulation to ensure steady-state results. Additionally, the effect of the return wages probability parameter on model dynamics was explored. This parameter represents the effectiveness of a laborer's report on decreasing theft propensity. It is hypothesized that a higher probability of reporting success will lead to greater punishment for the employer and a reduction in future theft propensity. Sensitivity analysis was performed by running simulations for reporting success rates of 10% to 50%, in addition to the default 1% rate. The simulation period of two years with 30 replications was used for all runs.

- *Output Validation:* Output validation is the process of comparing the model results to real-world data. However, due to limited real-world data available for validation of our ABM model, we instead verified the logical consistency of our model output against the initial conditions and the rules governing agent behavior. In our ABM model, we used empirical results from our case studies to validate the non-intervened nominal setting run of the model. To accomplish this, we simulated the model for two years with calibrated parameter values and replicated the simulation 30 times. We expect the output behavior to be consistent with current real-world conditions, in which workers remain uninformed about their rights and wage theft reporting does not occur. Over time, total wages stolen should increase as wage theft occurs and no wage thefts are reported. Based on our artifactual study of day laborers, we know that laborers probabilistically decline work from employers who have previously engaged in wage theft. As a result, we expect to observe the total number of jobs declined by laborers to increase over time as they experience more wage theft.

**Table 2.** ABM Parameters.

| Parameter | Value |
| --- | --- |
| Laborer population | 100 |
| Cost of living ($ per day)–min | $40 |
| Cost of living ($ per day)–max | $60 |
| Employer population | 50 |
| Job rate (per week) | 5 |
| Theft percent range–min | 10% |
| Theft percent range–max | 25% |
| Theft propensity | 0.2 |
| Job day range–min | 1 |
| Job day range–max | 1 |
| Job pay range ($ per day)–min | $80 |
| Job pay range ($ per day)–max | $150 |
| Job markup | 20% |
| Broadcast intervention–enabled | False |
| Broadcast intervention–campaign effectiveness | 0.015 |
| Broadcast intervention–adoption fraction | 0.015 |
| Broadcast intervention–contact rate (people/day) | 10 |

**Table 2.** *Cont.*

| Parameter | Value |
| --- | --- |
| Direct intervention–quantity | 0 |
| Direct intervention–adoption fraction | 0.015 |
| Direct intervention–contact rate (people/day) | 10 |
| Return wages probability | 0.01 |
| Punitive damages–nominal value | $500 |
| Punitive damages–probability | 0.01 |

Table 2 below presents the value or ranges used for each parameter in the ABM model. Most of these parameter values were determined based on insights gained from our empirical research, as described earlier, or from other relevant extant literature in the field, cited earlier. However, several assumptions are made that are supported anecdotally, and our assumptions are commented on next. The value of the job to the employer is assumed to be a 20% markup on their labor cost. Markup rates in practice vary extensively, ranging from a few percent on commodities to 100% on some highly technical services. We believe 20% is a reasonable rate for the type of work typically provided by day laborers. In the ABM the markup is used to calculate the value of the job to the employer and is factored into the Funds$_{total}$ value stored by an employer agent. This assumption has been held constant across our simulations. The three intervention parameters: campaign effectiveness, adoption fraction, and contact rate are based on an empirical study by Redmond of a campaign to create a behavior change, namely the cessation of smoking [57]. Like Redmond's study, we are examining prosocial behavioral changes rather than the increases in awareness and familiarity that are the focus of many other studies. We have included a review of empirical studies using the Bass diffusion model in our supplement for the interested reader. Here again, we held these parameter assumptions constant across our simulations.

We are assuming that whenever a report is made, there is a 1% probability of the report resulting in the successful restoration of stolen wages to a laborer. Our research on this subject corroborates other studies cited above. Basically, wage thefts are almost never reported through official channels and, when they are, positive outcomes almost never benefit laborers. Similarly, we assume a 1% likelihood that a successful wage theft report also includes a return of punitive damages along with the stolen wages to the laborer. We tested the sensitivity of these assumptions in our simulation experiments. Last, we assume the nominal value of the punitive damages to be $500. This value represents about a week's wages for a day laborer. It is worth noting that day laborers with whom we have spoken in our interview are not generally interested in receiving punitive damages. They have explained to us that they just want to be paid what they have earned. We hold this punitive damage assumption constant in our simulations, and its impact is limited stochastically only to cases of a report with a successful outcome and, given that success, to cases where such damages are assessed.

*3.8. Simulations*

Within this section, we present the results obtained from the experiments run for each component of the agent-based model validation framework. To allow the educational interventions to establish a stable state of worker rights awareness, the simulations were executed for a period of two years. The parameter settings were initialized according to Table 2, wherein the model was configured such that laborers found work for roughly 2.5 days per week while facing wage theft on approximately 20% of their working days, resulting in a loss of 20% of their earnings. These settings, which were observed during the artifactual study of day laborers seeking employment at a worker center, serve as the foundation for our output validation.

**Nominal**. Figure 5 shows that consistent with current conditions in the real world, wage theft occurs with a growing total for wages stolen, workers remain uninformed about their rights, and no wage thefts are reported. Consistent with our interviews, laborer agents can learn from the wage thefts they endure, probabilistically declining work from employers who have previously perpetrated wage theft against them. Based on our conversations with workers, the decision to decline is a function of the recency of the prior theft experience from the employer offering the job balanced against their current financial needs.

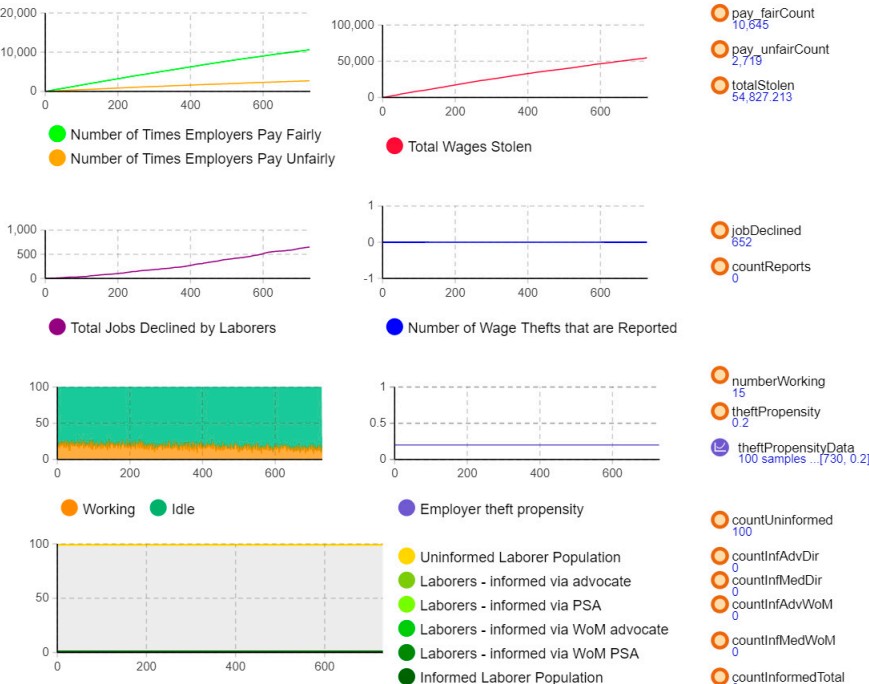

**Figure 5.** Non-Intervened Nominal Setting. Figure 5 displays the results of a two-year simulation at current, non-intervened settings, including $P_{success}$ = 0.01. The *X*-axis in all the graphs represents the simulation days.

**Adding Education**. Figure 6 demonstrates the educational benefits of two advocates working directly with laborers at the worker center to provide education about their rights as employees, which lead to a decline in the number of uninformed laborers over time. A growing number of wage thefts are reported; however, the probability of reporting success is low, and no significant reduction in the employer's theft propensity or total wages stolen is observed.

In Figure 7, the impact of adding a PSA campaign in addition to the two advocates as an intervention to the model is presented. The results show that the number of uninformed laborers declined faster than with just the advocates, and a steady state is reached earlier. The efficacy of the educational intervention is improved through the direct intervention complemented by the PSA campaign, and its effectiveness is dependent on the effectiveness of the PSA campaign. However, our model is using campaign effectiveness settings drawn from literature, and the benefits seen in the simulation are quite modest.

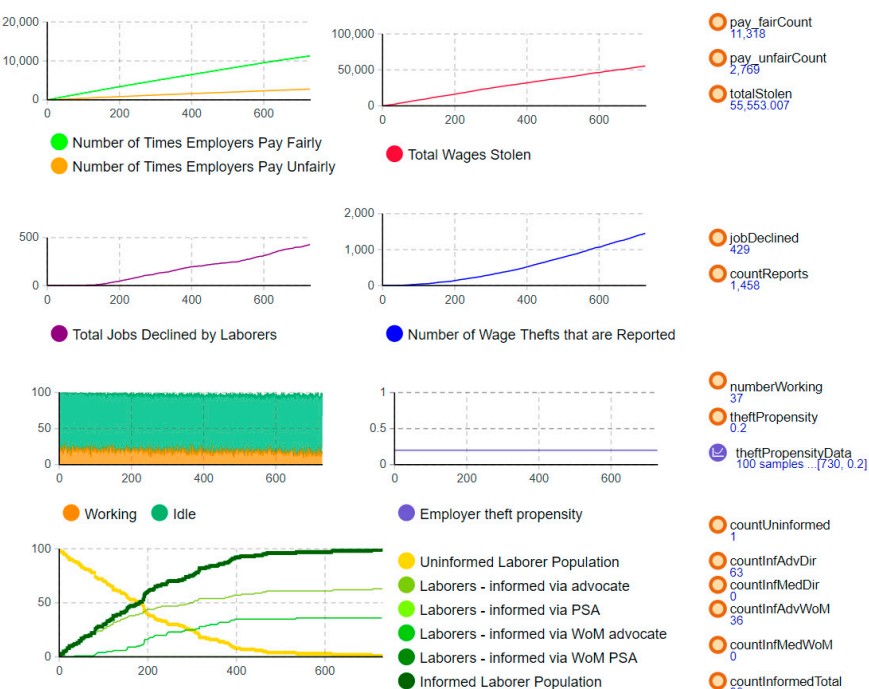

**Figure 6.** Direct Advocacy Intervention. Figure 6 displays the results of a two-year simulation with the deployment of two advocates who provide training about workers' rights, including $P_{success}$ = 0.01. The *X*-axis in all the graphs represents the simulation days.

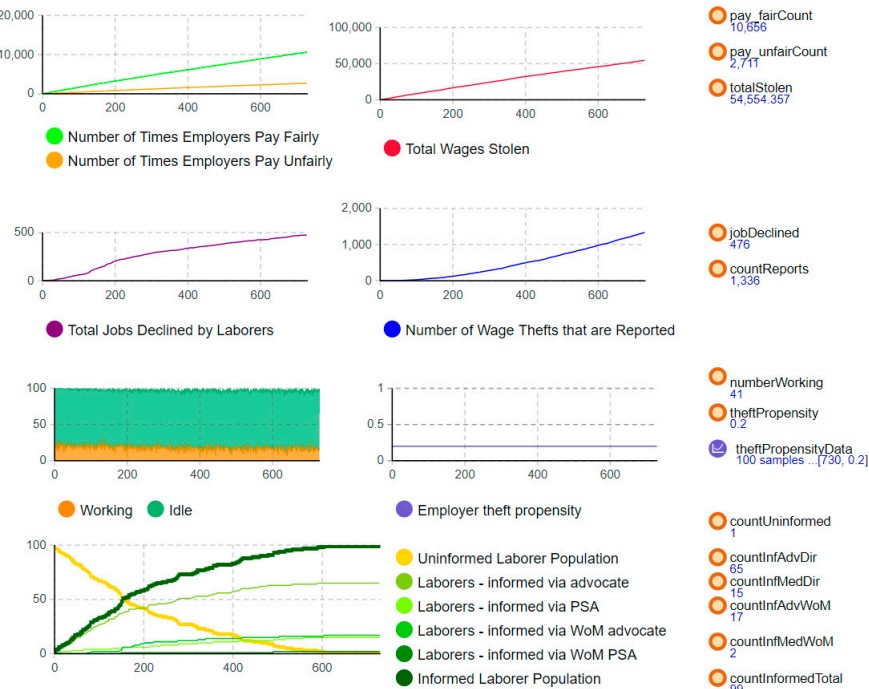

**Figure 7.** Direct Advocacy with a Complementary PSA. Figure 7 displays the results of a two-year simulation with the deployment of two advocates who provide training about workers' rights and a PSA, including $P_{success}$ = 0.01. The *X*-axis in all the graphs represents the simulation days.

In addition to providing for visual examination of effects, as shown in Figures 5–7, the ABM was used to generate samples of data analyzed separately. Following [56], we replicated all our simulations 30 times. Figure 8 presents the results obtained from these replications of the simulation to investigate the effects of two interventional strategies in comparison to the nominal state. The figure displays the theft propensity over time

by intervention and the probability of success ($P_{success}$) when reporting wage theft. The data from individual runs and an average of runs are shown in a linear model with a 0.999 confidence interval. Across the simulations, the synthetic dataset analyzed includes a total of 525,690 observations across the three interventional settings with $P_{success}$ set at the nominal, of current real world, value of 0.01. Figure 8 illustrates that the educational interventions do not create meaningful change in employers at $P_{success}$ = 0.01. Note that the vertical axis has been allowed to scale with the data, confirming that there are tiny, structurally consistent changes predicted by the model even at $P_{success}$ = 0.01. It is also apparent that the additional change induced through the inclusion of a PSA is small with the efficacy of typical campaigns that we assume from our literature review. These simulations suggest that in addition to providing education to laborers about their rights as workers, other changes may be required for education to translate into change among employers.

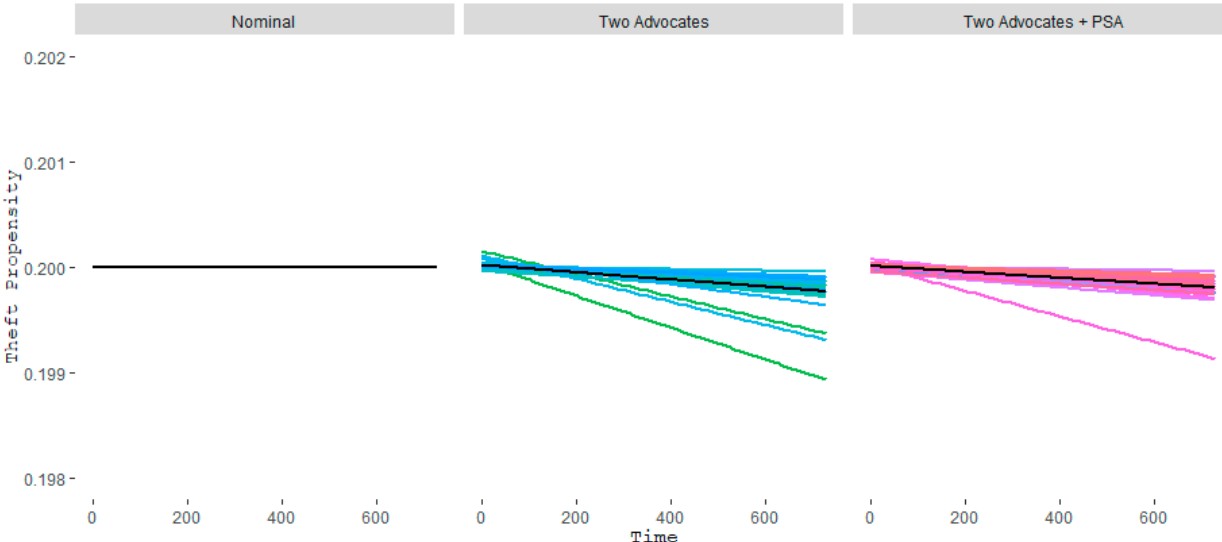

**Figure 8.** Theft Propensity over Time at $P_{success}$ = 0.01 by Intervention (30 Replications plus Average in Black).

**Examining other Structural Changes**. To that end, we continue this analysis by examining the change that might be possible if educational interventions are designed to be accompanied by other, unspecified system changes that allow for degrees of improvement in the likelihood of success when reporting occurs. For simplicity here, we limit this examination to the intervention scenario of two advocates supplemented by a PSA. We vary $P_{success}$ from the current, nominal, value of 0.01 up to 0.50.

Figure 9 illustrates the changes in system dynamics that were observed at the end of a two-year simulation period across six levels of probability of success for reporting wage theft ($P_{success}$), ranging from the present state (0.01) to five levels of enhancement (0.10 to 0.50). The simulation was replicated 30 times, and the summary analysis focused on the effect of increasing $P_{success}$ on the cumulative wages stolen, jobs declined, and thefts reported. Our findings indicate a modest negative association between $P_{success}$ and cumulative wages stolen, with employers committing less wage theft as the probability of punishment increases. Specifically, as wage theft declines, there is a corresponding modest reduction in both the number of reported thefts and jobs declined, as the employer behavior correction feedback loop mitigates the need for employees to decline jobs or report theft.

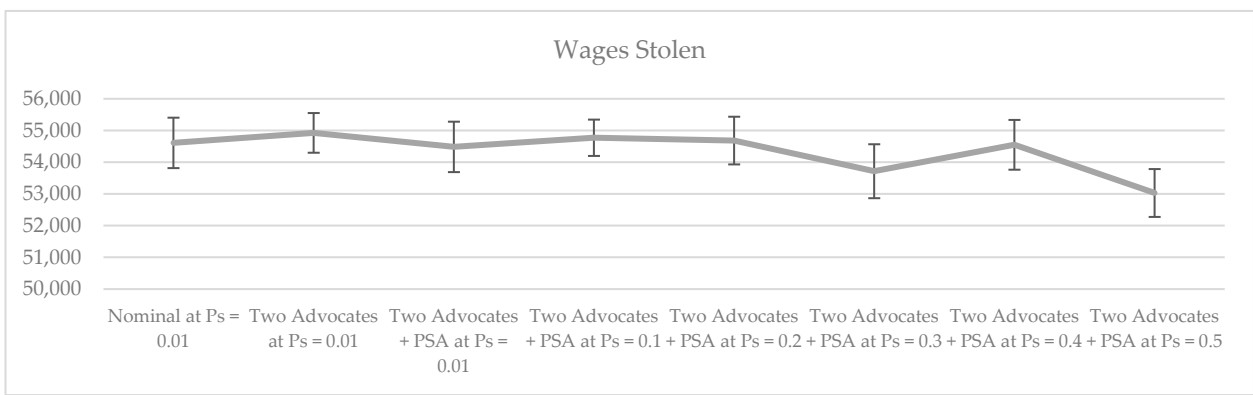

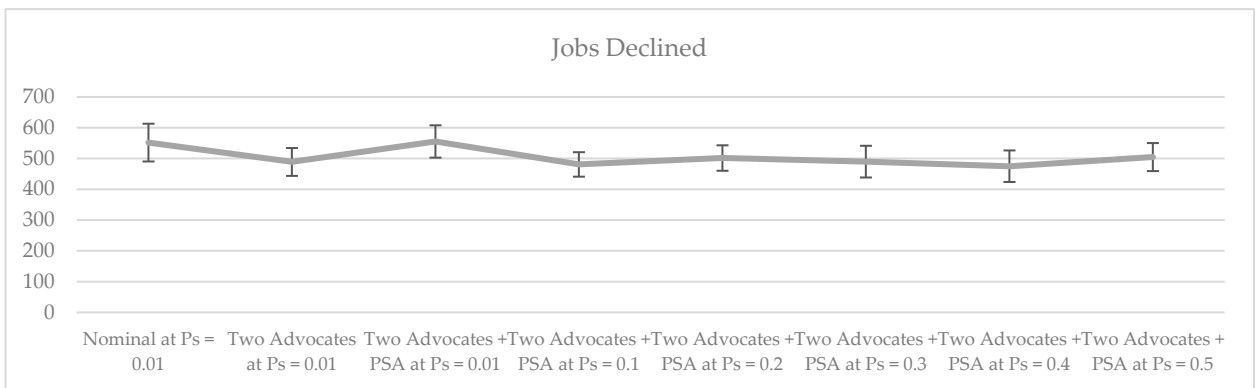

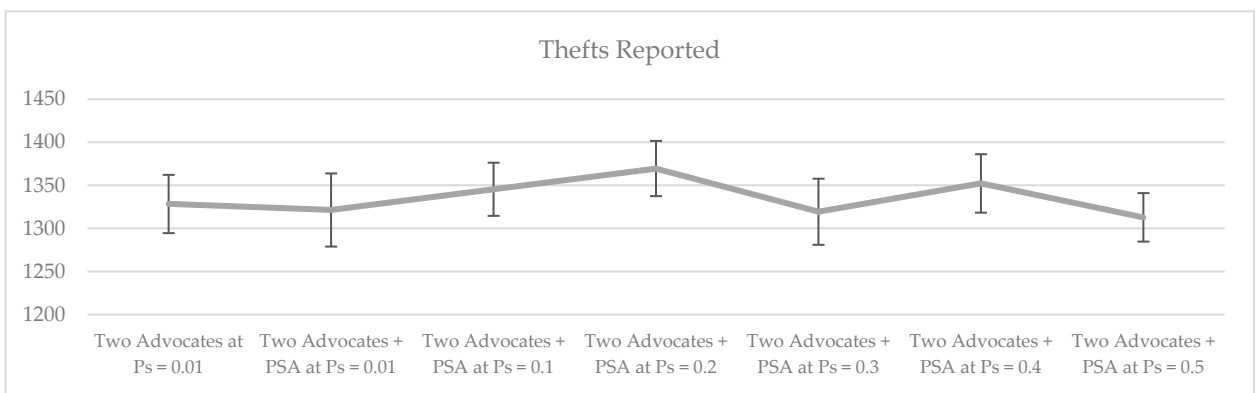

**Figure 9.** Select Cumulative Simulation Outputs at Two Years from 30 Replications (Average and 95% CI). Figure 9 summaries key metrics across the replicated simulation runs for the non-intervened system, the system with 2 advocates, and the system with 2 advocates supplemented by a PSA, the later with several levels of $P_{success}$. The runs highlight the changes in system dynamics at the end of the simulation horizon. All 30 replications were run for two years.

Similar to Figure 8, we have extended this analysis to include the theft propensity variable, which tracks employer behavior and influences decision-making in the ABM. Figure 10 superimposes the smoothed theft propensity plots for $P_{success}$ at 0.01, 0.1, 0.2, 0.3, 0.4, and 0.5. For completeness, the summary of the mixed linear model with a random intercept for replication is reported in Table 3, however, in this demonstration, the specific parameter estimates are of less importance than the structural relationships shown visually.

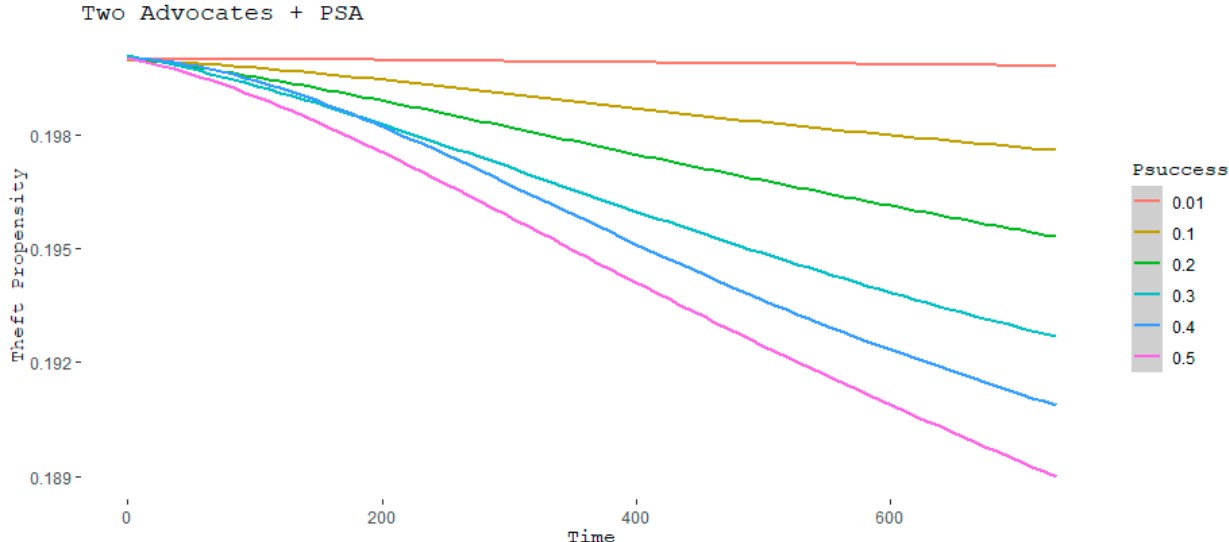

**Figure 10.** Theft Propensity over time for Two Advocates + PSA at various levels of P$_{success}$ (Results averaged over 30 replications).

**Table 3.** Linear Model Fixed Effect Summary.

|  | Estimate | Std. Error | t Value | Sig. |
|---|---|---|---|---|
| Intercept | $2.000 \times 10^{-1}$ | $7.658 \times 10^{-5}$ | 2611.19 | <0.000 |
| Time (days–linear) | $-2.177 \times 10^{-7}$ | $7.323 \times 10^{-9}$ | $-29.73$ | <0.000 |
| P$_{success}$ | $1.360 \times 10^{-3}$ | $1.020 \times 10^{-5}$ | 133.29 | <0.000 |
| Time $\times$ P$_{success}$ | $-3.293 \times 10^{-5}$ | 2.419 | $-1661.38$ | <0.000 |

Caption: A generalized linear mixed model with a random intercept for replication was used to estimate the effects of a range of values for P$_{success}$ over time (days). The model was fit in R using the lme4 package with Satterthwaite's method to produce t-test values with significance estimates.

Conceptually, this analysis illustrates that behavioral interventions like education may have a more profound impact when paired with system changes that allow such behavioral change for one agent (laborers) to translate into change for another agent (employers). This conjecture will be examined further in the discussion below. Further, this model indicates that the degree of improvement in reporting required to create meaningful change among employers is substantial and that the time horizon involved is measured in years.

As a demonstration of how ABM simulations might inform the next iteration of the proposed framework for developing interventions, we conclude our analysis with the following observations:

1. Using representative values for the dynamics of the diffusion of information, PSAs added only incremental benefits beyond the deployment of advocates. This is consistent with real-world practice, where PSAs are infrequently used.

2. Increased awareness of workers' rights and reporting options may need to be accompanied by other changes in this complex sociological system to allow the increased reporting to be effective. This is also consistent with what we heard from laborers, that among the few who had some awareness and familiarity of reporting options, perceptions of efficacy are extremely low. In fact, informal mechanisms appear to be preferred over formal reporting.

3. The duration of educational programs needs to be in years to achieve full effectiveness. To be fair, we utilized a BDM driven by parameters drawn from other real-world settings, but more needs to be learned about how information is shared by laborers.

4. Our ABM was designed based on and calibrated by the results from three artifactual experiments among laborers. Much was revealed about laborers' lived experiences

in these experiments, but more needs to be learned in the next iteration about the decision-making processes of employers and about the specific interpersonal dynamics between laborers and employers.

## 4. Discussion

Consistent with the original process developed by Battista and colleagues, the present study has illustrated the potential of steps 1–3 in the process outlined above. This approach can add meaningful learning and calls for additional iterations before advancing to subsequent steps. Specifically, the case study presented here illustrated the application of the framework to develop interventions to disrupt illicit behavior and enhance prosocial behavior in complex sociological systems and applied this framework to the problem domain of labor exploitation and trafficking. Steps 1 and 2 provided evidence to propose a series of hypothetical interventions. The first iteration of Step 3 tested two of those interventional candidates in a virtual (ABM) setting. Our case study has shown the utility of connecting relevant theories (Step 1) and has outlined a plan for the next round of artifactual experiments (Step 2) and model enhancement (Step 3). Application of Steps 4 through 8, and iterations therein, remain future work.

The interventional scenarios included in the current case study illustrate the potential efficacy of educational campaigns in two forms, direct advocacy, and PSAs. This potential is examined in the case study as conditional on the degree of change in the likelihood of a successful outcome from reporting exploitation. Advocates can effectively make outreach and transfer information that will continue to spread throughout the community through word of mouth. Worker center staff and advocates from other allied service providers are an effective way to deliver such training to the laborer community. PSAs provide a mechanism for extending advocate outreach. More needs to be learned about the differential efficacy of advocates and PSAs and the degree to which these strategies create equity for laborers. Our case study suggests that education alone is insufficient to eliminate wage theft. The likelihood of a successful outcome from reporting will need to increase dramatically before substantive system change will be seen. Further, this case study suggests substantive system change will require sustained effort (2 years in our illustrative model). Our example case study employed reasonable levels of resource allocation and campaign effectiveness. Additional empirical research and modeling are required to move our work from an illustrative case study to a real-world intervention design project.

Specific to methodology, an agent-based model was developed as part of the extension of the framework. While simplistic, this model illustrates the utility of the integration of decision and behavioral sciences because the approach allows early interventional candidates to be examined in a virtual environment before deployment in the field, including pilot deployments. As an example, the critical thinking process utilized during the development of the simulation environment elucidated additional information needed to expand the functionality of the ABM to endow employer agents more completely with appropriate behaviors in the next iteration of the intervention design process. Our reflective analysis of the current study suggests that the next steps for this research should include empirical investigations of perpetration behavior grounded in theory and the incorporation of network-based data structures and data that allow the model to address the interactions between perpetrators and victims. These improved measurements will allow future research to incorporate network models that examine these interactions directly, compared to the simple Bass Diffusion Model employed in this initial iteration of the framework.

Further, a simplified diffusion model (the BDM) was used in this case study to model how information flows from advocates and PSAs to and among workers. Additional data collection before and after deployment of an educational intervention would support a more detailed and dynamic network modeling approach, e.g., stochastic actor-oriented models. Such stochastic actor-oriented models (SAOMs) [58,59] have the potential to elucidate the social network ties between community members, employers, and community members and employers. This modeling activity can improve understanding of the respective and

interactive effects of influence, selection, and social norms in accounting for ecosystem participants' co-evolution of fair and exploitive behaviors. SAOMs can also be formulated to include the possible diffusion of interventions into the network [58,60]. Such models require different data collection procedures than those used in the present study, including data that records the network structure and dynamics between network participants. An improved understanding of the interactions between ecosystem participants would support improvement to the ABM test bed and further preparation for the testing of interventions at scale in Step 8.

To effectively address labor exploitation, we must better understand the motivations and actions of perpetrators and victims. One of the most well-known criminal behavior theories, the theory of differential association, attests that criminal behavior is, at its core, learned behavior [61]. This learning theory states that the process of learning criminal behavior occurs through an ongoing association with those who display criminal attitudes or values. Learning theory, the overarching framework from which differential association theory came, is based on the concept of conditioning, where behavior is related to its relative environment. Operant behaviors are one step further, wherein behaviors occur in the presence of a specific environmental or stimulus and can be maintained by the environment's response to the specific behavior. This maintenance of a particular behavior is amplified when the response is positive. Criminal behavior, like all social interactions, is operant behavior. As such, the behavior is maintained by the responses it brings up in the environment. Specifically, the frequency of the behavior is determined by the consequence or lack thereof. The criminal act of underpaying employees can produce costs saved and more profit, which would fall under a positive stimulus and provide a reinforcement to repeat the criminal action. Sutherland's theory also states that fellow human beings often act as the primary reinforcers of criminal activities [61]. If exploited day laborers do not voice their protests, the employer may believe that their actions are not causing significant enough harm to stop. Likewise, suppose other employers see that one can extract more labor for less and comment on it, whether from envy, admiration, or simply an observation. In that case, the exploitative employer may be encouraged to continue with the criminal action.

As a familiar foundation, Merton's strain theory states that the social structure and environment, rather than culture or company, pressure specific individuals to commit crimes [62]. This theory was initially developed to explain the high frequency of crime in low-income areas instead of in high-income areas. Various types of strain–stress, financial distress, perceived disrespect, desired status, and dissatisfaction–can lead an individual to engage in specific criminal activity. While these strains are present in various socioeconomic levels, strain theory proposes that individual strain is more severe and frequent among people who desire and internalize the societal goals of high status and wealth but are aware of the barriers that exist for their attainment [62]. This clash between expectations and perceived attainment can drive people to engage in criminal activity out of desperation or a twisted sense of justification, in which they feel the system at large, not themselves, is to blame for their actions. Individuals who desire financial success, but view going to college as unattainable, are more likely to seek other avenues to achieve that success, including crime. For employers who employ day laborers, exploiting their workers may relieve some of the strain they experience. Analyzing the demographics of employers in this informal sector, specifically looking at education level and economic aspirations could reveal strains that would increase the likelihood of offending.

Such future research on perpetration behaviors will need to address enduring challenges associated with measuring those behaviors. For example, the authors have investigated perpetration behavior in sexual violence and misconduct. The most common strategy for measuring perpetration that we are aware of includes modifying established victimization protocols designed to measure specific behaviors, e.g., the Sexual Experiences Survey (SES) [63], into perpetration-focused behavioral protocols. These modifications have been reliable, but scholars believe they likely underrepresent perpetration rates [64–66]. Measurement tools for human trafficking are available, but fewer studies have been con-

ducted using them to examine perpetration. The Trafficking Victim Identification Tool (TVIT) screener [67] was designed for labor and sex trafficking. The Human Trafficking Interview and Assessment Measure (HTIAM-14) and the Human Trafficking Screening Tool (HTST) are rigorously designed screening tools that include labor exploitation. Still, they were developed primarily to identify sex trafficking [68,69]. Our research has largely been informed by methods developed by Zhang and colleagues, whose research is specific to labor exploitation and trafficking, focusing on victimization [70]. Our future research will examine how to adapt such measurement approaches best to identify perpetration.

The current study suggests the potential of a third class of interventions, restorative processes. Restorative justice focuses on a healing process for victims of crimes. Restorative processes involve balancing the needs of individuals and communities in the pursuit of justice [9]. Rather than focusing solely on punitive justice, restorative justice seeks to raise the understanding and impact of crime for those who have caused harm. Traditional justice methods can prove challenging for victims of labor trafficking; in the United States, day laborers are often foreign-born or primarily speak Spanish. These are barriers to interacting with law enforcement. Studies have found that while 52% of survivors are referred to the criminal justice system to address their cases, most are not interested in pursuing specific criminal justice solutions against their traffickers [9,71]. A combination of mistrust and negative experiences with law enforcement leaves many victims of wage theft unsure of how to seek justice. Furthermore, human trafficking victims see the prevention of harm to others rather than a narrow focus on the incarceration of their trafficker as an ideal endpoint in seeking justice [71]. Survivors have voiced critiques over the effectiveness of incarceration in promoting accountability and changing behavior [11]. Foreign and natural-born survivors of human trafficking view justice as two-tiered: receiving their stolen wages and preventing the traffickers from continuing to harm others [71]. Foreign-born survivors were particularly adamant about the immigration status of perpetrators, stating that "...traffickers should be prevented from re-entering the US and obtaining permission to hire additional workers" [72]. While the exact number of perpetrators who hold illegal immigration status is unknown, the high frequency of survivors mentioning the immigration status of employers presents an opportunity for a connection between the victims and perpetrators [72]. Day laborers have expressed confusion about the criminal justice system and their rights as workers, so similar confusion may exist for foreign-born employers [10]. To understand its occurrence, addressing labor trafficking must factor in the reasons for the criminal behavior [73–75]. Restorative justice initiatives recognize the humanity and complexities within the victim and preparator alike and seek justice by working with community members to heal and ultimately change their behavior [10,71,72].

Although, restorative processes were not empirically examined evidence gathered in this case study supports the potential of this approach, especially in the context of a labor center wherein laborers, employers, and center staff convene in a venue that is amenable to restorative processes to disrupt illicit behavior and build trust and open communication. The current study illustrates the limitations of education without a focus on improved outcomes. Such system changes can be implemented in formal reporting channels. But system changes can also be implemented informally, for example, at a worker center. How might such changes be achieved?

The type of intervention best suited to address labor exploitation should consider participants' desired outcomes within the social system. Interviews with day laborers have shown that the majority believe 'money recovery' to be the most successful outcome, as opposed to the exploitative employer facing legal ramifications beyond wage recovery [2,71]. Many day laborers would prefer to pursue justice by correcting a past grievance without the process and ramifications of involving the legal and criminal justice system [66,72]. Additional analysis of the data reported by the authors has revealed a willingness from day laborers to reengage with previously exploitative employers under certain conditions [2]. This initial inclination to work once more with an offender could safely be supported through restorative justice programs [9]. Interventions that center on restorative justice may

allow victims to safely voice their needs and establish norms for individual and community healing and offender accountability. In this way, such restorative approaches to employment can foster cooperation in the sense of Fiske [76] and contribute to sustainable economic growth and well-being in the sense of Coscieme et al. [77] and Fioramonti et al. [78].

Differential association and strain theories recognize that perpetration is learned behavior grounded in contextual stressors. Various forms of justice have been applied historically, with a common focus broadly in labor exploitation and trafficking being on procedural strategies of reporting infractions with penalties for offenders. Coupled with common mistrust by victims of formal reporting mechanisms and law enforcement, restorative processes have potential, especially within a worker center. The same education tools used for day laborers could be applied to employers as a preventative measure against exploitation. Just as advocates work with day laborers to understand their rights and reduce their tolerance for being exploited, so can employers learn from and work with day laborers. Fostering empathy, justice, and understanding of why wage theft and other forms of labor trafficking are wrong, beyond being a crime has significant value.

## 5. Conclusions

The cumulative evidence from our artifactual and virtual experiments to date supports continued research into the response of employers to interventions designed to protect day laborers and the potential of restorative processes applied within a labor center as a means of addressing exploitative behaviors, like wage theft.

We have extended the process by Battista and colleagues [1] for developing an intervention to alter behavior with the introduction of agent-based modeling. Additional steps (3, 5, and 7) illustrate where simulation models can be incorporated to improve the understanding of the problem domain and generate data for evaluating the effects of potential interventions.

1.  Gain an in-depth understanding of the community and context; it is possible to identify relevant actors, types of problematic behaviors, and potential drivers.
2.  Develop hypothetical interventions based on testable hypotheses about the drivers of the problem behavior.
3.  Build simulation models that capture the ecosystem dynamics and structure of the interventions.
4.  Experimentally test hypothetical interventions.
5.  Explore the impact of hypothetical interventions with models in the virtual simulation laboratory before (or in parallel) real-world pilots.
6.  Pilot interventions based on the mechanisms identified.
7.  Update models and analysis to optimize plans for testing interventions at scale.
8.  Scale-up tested interventions and set up systems to monitor, evaluate, and adjust.

A case study of labor exploitation and trafficking demonstrated the utility of developing and testing interventions using participatory methods and simulation and modeling methods to learn more about the system virtually to reduce risk and time when testing or deploying those interventions in the real-world system. Specifically, worker centers are viable venues for various interventions that target the disruption of illicit behaviors like wage theft and promote prosocial behaviors. For example, education targeted at laborers and employers about workers' rights and the obligations of employers to workers is a critically needed primordial prevention. Such an intervention would benefit from the inclusion of labor organization principles optimized for the informal setting of day labor. Additionally, education should include curricula on workplace safety, the second most common form of labor exploitation. Such multilayered system approaches have been recognized previously as viable for a range of policy and social action domains, including sexual violence and misconduct [79,80], sex trafficking and sexual exploitation [81,82], and wildlife trafficking and emerging infectious disease [5], among others.

The three added decision science phases of activity in the behavioral science process are constructed from data from local communities to reflect the ground truth and enhance

the accuracy of modeling and analysis. Labor exploitation [6] can be considered a complex system that is ultimately rooted in human behavior, with direct and indirect bidirectional effects across a network of employees and workers. Approaching anti-exploitation interventions through a multidisciplinary lens can lead to a greater understanding of the tolerance, willingness, and behavior of exploitation and create a framework for developing effective strategies for change. Agent-based simulation is an appropriate methodological approach to include in this process because such models can reveal emergent behavior in more complex real-world modeling environments and generate data that can be used in other models, e.g., optimization models and reinforcement learning methods, which, in turn, can be used to plan and deploy interventions to meet policy objectives.

**Supplementary Materials:** The following supporting information can be downloaded at: https://www.mdpi.com/article/10.3390/soc13040096/s1.

**Author Contributions:** Conceptualization, M.K.-K. and N.B.-A.; Data curation, M.Y.-P.; Formal analysis, M.Y.-P., N.V. and K.T.; Funding acquisition, M.K.-K.; Methodology, M.K.-K.; Project administration, M.K.-K.; Software, N.V.; Supervision, M.K.-K. and K.T.; Validation, N.B.-A.; Writing—original draft, M.K.-K., M.Y.-P., N.V., K.T. and N.B.-A. All authors have read and agreed to the published version of the manuscript.

**Funding:** This research was funded by the National Science Foundation Grants D-ISN: TRACK 1: Collaborative Research: Disrupting Exploitation and Trafficking in Labor Supply Networks: Convergence of Behavioral and Decision Science to Design Interventions (Award Number 2039983) and EAGER: ISN: Disrupting Exploitation and Trafficking Labor Supply Networks in Post-Harvey Rebuild (Award Number 1838039). The first four authors also receive broad institutional support as part of a stream of research at the IC$^2$ Institute investigating the merits of a well-being economy.

**Institutional Review Board Statement:** This study was reviewed and approved by the Institutional Review Board of the lead author's institution, protocol number 2018060067, approved 15 October 2019.

**Informed Consent Statement:** Informed consent was obtained from all subjects involved in the study.

**Data Availability Statement:** The ABM model presented in the manuscript is available from the corresponding author. All data used to inform that model were described in (2).

**Acknowledgments:** The paper is the result of a larger research stream that involves a collaboration with several additional researchers. We acknowledge the contributions of our other team members devoted to the disruption and remediation of human trafficking and exploitation, including: Bruce Kellison, Melissa I. M. Torres, Dixie Hairston, MacKenna Tally, Daniel Lazcano.

**Conflicts of Interest:** The authors declare no conflict of interest.

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
