# Peer review of "A Framework to Develop Interventions to Address Labor Exploitation and Trafficking: Integration of Behavioral and Decision Science within a Case Study of Day Laborers"

_societies, doi:10.3390/soc13040096_

Round 1

Reviewer 1 Report

See attached

Author Response

Thank you for your thorough review of our manuscript.  We have endeavored to address your feedback completely, and provide a detailed response as an attachment.

Reviewer 2 Report

I appreciate the chance to read this manuscript.

Some comments/suggestions below.

About formal presentation:

- the manuscript presents some typos (few), to be corrected. For example in the Abstract "being economy...";

- the theme to develop a framework that "integrates behavioral and decision science methods to design and evaluate interventions to disrupt illicit behaviors" is extremely pertinent and complex;

- there are several keywords that join more than one word. I think this is unnecessary in: "Labor exploitation and trafficking", "intervention design framework";

In terms of content:

- very good framework in the corpus of knowledge on "behavioural framework," as well as description of the studies carried out and their internal articulation;

- rewrite for better readability "2. A Framework to Develop Interventions Targeting Human Behavior 232 Situational crime prevention works within a 5-step intervention framework: 1) 233 increasing the risk of the criminal action, 2) increasing the effort of the criminal action, 3) 234 reducing the reward of the criminal action, 4) reducing the provocations, and 5) removing 235 the excuses (26). Within this framework exist specific techniques which can be applied 236 depending on the type of criminal activity. A systemic review of situational crime 237 prevention measures revealed the most and least used in preventing undesirable criminal 238 behavior. The most common intervention methods were increasing risk and increasing 239 the effort of criminal acts. The framework has been used in studies from various 240 disciplines and industries. 241" and remove the underlining;

- excellent Figure 4 and 5 but where are Figure 2 and 3?

- good explanation of the potential but also of the limitations of the proposal: "While simplistic, this model illustrates the utility of the 952 integration of decision and behavioral sciences because the approach allows early 953 interventional candidates to be examined in a virtual environment before deployment in 954 the field, including pilot deployments.. . .".

Note: I couldn't open "Externally hosted supplementary files Supplementary 1 Link: https://utexas.box.com/s/65s8uhvtrr1c73qew0i2a02xvlxe6z06".

Author Response

(The authors gave the same response as above.)
